# Rottlerin inhibits macropinocytosis of Porcine Reproductive and Respiratory Syndrome Virus through the PKCδ-Cofilin signaling pathway

Yeonglim Kang[1], Jong-Chul Choi[1], Joong-Bok Lee[1], Seung-Yong Park[1], Changin Oh[2]*

1 Laboratory of Infectious Diseases, College of Veterinary Medicine, Konkuk University, Seoul, Republic of Korea, 2 Department of Genetics, Yale University School of Medicine, New Haven, Connecticut, United States of America

* changin.oh@yale.edu

## Abstract

Rottlerin exerts antiviral activity against various enveloped viruses, yet the mechanism by which it inhibits viral replication and the associated signaling pathways remains unclear. Here, we investigated the mechanisms for the antiviral effects of rottlerin against Porcine Reproductive and Respiratory Syndrome Virus (PRRSV) in vitro. We demonstrate that PRRSV enters host cells via macropinocytosis. Rottlerin, a PKCδ inhibitor, partially inhibits PRRSV entry by decreasing actin polymerization, as evidenced by alterations in actin dynamics. LIM domain kinase 1 (LIMK1) is essential for PRRSV replication, and PRRSV infection-induced cofilin activation, which was reversed by rottlerin treatment. Our findings suggest that a subset of PRRSV utilizes PKCδ-mediated actin dynamics to enter cells via macropinocytosis, and that rottlerin is a potential antiviral molecule targeting this entry pathway.

## Introduction

Macropinocytosis is a form of endocytosis that facilitates the uptake of fluids, extracellular particles, and membranes into large intracellular vesicles [1]. Activation of actin and microfilaments induces macropinocytosis through the formation of lamellipodia, membrane ruffles, and blebs on the cellular periphery [2]. Various viral families, such as human immunodeficiency virus (HIV), vaccinia virus, and adenovirus, and various types of bacteria take advantage of macropinocytosis as a direct means of cellular internalization or an indirect mechanism to assist the penetration of particles that have entered by other forms of endocytosis [2]. Thus, blocking macropinocytosis is a potential strategy for controlling broad-spectrum viral and bacterial infections.

Porcine Reproductive and Respiratory Syndrome Virus (PRRSV) is an enveloped positive-sense ssRNA virus belonging to the family Arteriviridae and order Nidovirales [3,4]. PRRSV infection causes reproductive impairments in sows and respiratory

**Data availability statement:** The minimal data set is included within the manuscript and its supporting information files.

**Funding:** This research was supported by the Korea Institute of Planning and Evaluation for Technology in Food, Agriculture, and Forestry (IPET) through the Agriculture, Food, and Rural Affairs Convergence Technologies Program for Educating the Creative Global Leader Program, funded by the Ministry of Agriculture, Food, and Rural Affairs (MAFRA) (320005-04)." There was no additional external funding received for this study.

**Competing interests:** The authors have declared that no competing interests exist.

diseases in pigs of all ages, resulting in a critical economic impact on the global swine industry [5,6]. It has been reported that PRRSV enters into host cells via low pH-dependent, clathrin-mediated endocytosis, utilizing CD163 as an essential receptor [7–9]. A recent paper claims that PRRSV uses macropinocytosis as an alternative entry route [10]. However, additional evidence and detailed mechanisms underlying PRRSV-induced macropinocytosis are necessary to support this novel finding. We previously demonstrated that rottlerin blocks PRRSV endocytosis by inhibiting PRRSV-induced protein kinase C delta (PKCδ) phosphorylation [11], suggesting that inhibition of PKCδ phosphorylation suppresses one or more PRRSV endocytosis routes. Given the crucial role of the PKCδ on PKCδ-cofilin signaling axis in regulating actin dynamics, which is essential for various cellular processes [12], we focused on investigating the effects of rottlerin on virus entry. Upon activation, PKCδ phosphorylates and activates LIM domain kinase 1 (LIMK1), which subsequently phosphorylates cofilin at Ser3, resulting in its inactivation [13,14]. This inactivation leads to the stabilization of actin filaments and the suppression of actin dynamics, which ultimately impacts cellular processes, including viral entry. This signaling cascade highlights how viruses exploit host cell signaling pathways to facilitate their internalization, positioning the PKCδ-cofilin pathway as a potential therapeutic target for antiviral strategies. Thus, we investigated the antiviral mechanisms of rottlerin against PRRSV, especially how rottlerin blocks PRRSV endocytosis to fully understand its antiviral potential and to explore its clinical applicability as a therapeutic option for PRRSV and other viral infections.

Here, we demonstrated that PRRSV enters MARC-145 cells via macropinocytosis during infection, which is inhibited by rottlerin treatment, by decreasing actin polymerization using fluorescence microscope. PRRSV macropinocytosis is mediated by the PKCδ-cofilin pathway and LIM domain kinase 1 (LIMK1), as indicated by the phosphorylation of cofilin and LIMK1 in western blotting. The results of the present study provide a possible explanation for the antiviral effect of rottlerin on other viruses and contribute to an understanding of the PRRSV entry mechanism.

## Materials and methods

### Virus infection

The PRRSV FL12 strain used in this study was rescued from the pCMV-FL12/24 plasmid that was kindly provided by Professor Won-Il Kim. Briefly, semi-confluent MARC-145 cells, a clone of the African monkey kidney MA-104 cell line [15], were transfected with pCMV-FL12/24 [16] using the ViaFect reagent (Promega, Madison, WI, USA) in a 1:3 ratio. After 72 hours of transfection, 10 ml of cell supernatants were harvested and passaged into MARC-145 cells. Virus titers were measured using quantitative reverse transcription–polymerase chain reaction (qRT-PCR) using Light-Cycler 96 (Roche, Switzerland) and titration using the Reed-Muench method [17]. Viral stocks were stored at −80 °C. Viruses were concentrated by centrifugation using a Beckman SW-41 Ti rotor at 160,000 × $g$ at 4 °C for 4 hours through a discontinuous sucrose gradient consisting of 1 mL of 25%, 35%, 45%, 55%, and 65% sucrose. The 35%–45% fraction was dialyzed and titrated for future experiments.

## Immunofluorescence assay and confocal microscope

Approximately $2.4 \times 10^4$ MARC-145 cells per well were seeded on an eight-chamber Flux slide (SPL, South Korea). On the following day, MARC-145 cells were preincubated with full-length rottlerin (5 µM) or 5-[N-ethyl-N-isopropyl] amiloride (EIPA) (50 µM) (Tocris, Bristol, UK) in $CO_2$ incubator at 37 °C for 2 hours or left untreated. For the viral endocytosis assay, cells were infected with PRRSV FL12 at a multiplicity of infection (MOI) of 50 for 1 hour. Cells were washed and treated with proteinase K to remove extracellular viruses for 45 minutes at 4°C. For the dextran uptake assay, cells were inoculated with CF®640R dextran (10,000 MW) (Biotium, Fremont, CA, USA) for 1 hour and subsequently observed. In the following step, cells were inoculated with both CF®640R dextran and PRRSV FL12 (MOI = 50), incubated for 1 hour, and then washed with 3 M sodium acetate buffer (pH 5.2). The cells were fixed with 4% paraformaldehyde for 15 minutes and permeabilized with Tris-buffered saline (TBS) containing 0.2% Triton X-100 at room temperature for 10 minutes. The cells were then probed as previously described [18] with minor modifications. CF®640R phalloidin (Biotium, Fremont, CA, USA) was used to label the F-actin for the visualization of actin polymerization. An anti-PRRSV nucleocapsid (N) monoclonal antibody (1:200) (Median Diagnostics, Chuncheon, South Korea) was applied to the cells and incubated for 1 hour to label the viral particles. The cells were scanned using an LSM 900 with Airyscan 2 (Zeiss, Oberkochen, Germany). The images were processed, and the corrected total cell fluorescence was quantified using ImageJ software 1.54j version (NIH, Bethesda, MD, USA).

## RNA interference

To silence PKCδ, LIM domain kinase 1 (LIMK1), and cofilin expression, double-stranded siRNA oligonucleotides were synthesized from Genepharma (China) as follows: PKCδ, 5′-CCAUGAGUUUAUCGCCACC-3′-dTdT; LIMK1, 5′-GAGC CCAGAUGUGAAGAAU-3′-dTdT; cofilin, 5′-CCUCUAUGAUGCAACCUAU-3′-dTdT; scrambled, 5′-CCUACGCCAAUUUC GU-3′-dTdT. MARC-145 cells were transfected with small interfering RNAs (siRNAs) targeting PKCδ, LIMK1, and cofilin, respectively [19] using Lipofectamine 3000 (Thermo Fisher Scientific, MA, USA) according to the manufacturer's instructions. Briefly, 10 nM of each siRNAs were diluted in Opti-MEM (Thermo Fisher Scientific, MA, USA), and Lipofectamine 3000 was separately diluted in Opti-MEM. These solutions were combined and incubated for 20 minutes to form the transfection complex, which was then added to 70% confluent MARC-145 cells and incubated at 37 °C for 24 hours.

## Western blot

To test the effects of rottlerin on the phosphorylation of LIMK1 and cofilin, confluent MARC-145 cells were treated with rottlerin (5 µM) for 2 hours at 37 °C. The cells were inoculated with PRRSV FL12 (MOI = 1) and incubated for 5, 15, 30, 60, 90 and 120 min. Cells were lysed using mammalian protein extraction reagent (Thermo Fisher Scientific) with a Halt protease phosphatase inhibitor cocktail (Thermo Fisher Scientific). Cell lysates were resolved using SDS-PAGE (Bolt 4%–12% Bis-Tris gel; Invitrogen, Waltham, MA, USA) and transferred onto nitrocellulose membranes (Komabiotech, Seoul, South Korea). The membranes were blocked with 5% skim milk in TBS with Tween 20 (TBS-T) for 30 minutes at room temperature and subsequently incubated with the following primary antibody: rabbit anti-cofilin (P-cofilin, Ser3) monoclonal antibody (1:1,000) (Cell Signaling Technology, Danvers, MA, USA), mouse anti-cofilin monoclonal antibody–horseradish peroxidase (HRP) (1:1,000) (Santa Cruz Biotechnology, Dallas, TX, USA), rabbit anti-LIMK1/2 (P-LIMK1, Thr508/505) polyclonal antibody (1:1,000) (BioVision, Waltham, MA, USA), rabbit anti-LIMK1 polyclonal antibody (1:1,000) (Cell Signaling Technology), or mouse anti-β-actin monoclonal antibody-HRP (1:50,000) (Abcam, Cambridge, UK) overnight at 4 °C. After five washes with TBS-T, the membranes were incubated with HRP-conjugated anti-rabbit IgG (1:5,000) (Komabiotech) for 1 hour at room temperature. Proteins were visualized using Clarity Western ECL Substrate (Bio-Rad, Hercules, CA, USA) and detected using FUSION Solo S (Vilber, Collégien, France). The density of the protein bands was measured using ImageJ software (NIH) after subtracting the density of the β-actin bands.

### qRT-PCR and virus titration

Viral RNA was extracted from the supernatants of PRRSV-infected cells using an RNeasy Mini Kit (QIAGEN, Hilden, Germany), according to the manufacturer's recommendations. Viral RNA levels were quantified using the RNA UltraSense One-Step Quantitative RT-PCR System (Invitrogen) with the LightCycler 96 System (Roche, Basel, Switzerland), as previously described [20]. The measured cycle threshold (CT) values were converted to RNA copy numbers and tissue culture infective dose 50% ($TCID_{50}$) equivalent/mL using a standard curve generated from CT values plotted against the corresponding $TCID_{50}$/mL. The viral titer ($TCID_{50}$/mL) of the collected samples was measured using the Reed–Muench method, as previously described [21].

### Statistical analyses

All data were analyzed using IBM SPSS Statistics 25 (IBM Corp., Armonk, NY, USA) and GraphPad Prism 6 (GraphPad Software, San Diego, CA, USA) using a nonparametric model (Kruskal–Wallis followed by Mann–Whitney). All data are presented as the mean ± standard deviation, and *, $p$-value <0.0332; **, $p$-value <0.0021; ***, $p$-value <0.0002; ****, $p$-value <0.0001 were considered significant.

## Results

### Rottlerin inhibits PRRSV endocytosis in MARC-145 cells

The precise mechanism underlying its antiviral activity and the role of other PKCδ-related cellular molecules remain unclear although it has been reported that rottlerin inhibits PRRSV endocytosis [11]. To investigate the role of the PKCδ and elucidate the antiviral mechanism of rottlerin during PRRSV infection, we examined PRRSV replication in the presence of rottlerin in MARC-145 cells. The cells were pre-treated with rottlerin (5 μM) or EIPA (50 μM), or subjected to PKCδ silencing, followed by infection with PRRSV FL12 (MOI = 1). Subsequently, viral replication was evaluated using three distinct parameters at 48 hours post-infection (hpi): PRRSV RNA synthesis, PRRSV nucleocapsid (N) protein expression, and infectious virus production. Consistent with our previous report, treatment with both rottlerin and siPKCδ significantly inhibited viral RNA synthesis at 2 days post-infection (Fig 1A), to a degree comparable to that of EIPA, when compared to DMSO-treated cells. Furthermore, the expression of the PRRSV nucleocapsid (N) protein (Fig 1B) and the production of infectious progeny viruses (Fig 1C) were notably reduced in both rottlerin and siPKCδ-treated cells, at levels similar to those observed with EIPA, in comparison to the DMSO-treated controls.

To further determine whether rottlerin blocks PRRSV endocytosis, we used immunofluorescence to observe the localization of PRRSV particles in infected MARC-145 cells after treatment with rottlerin or EIPA. The cells were treated with proteinase K to eliminate the extracellular viruses. As expected, confocal microscopy images of dimethyl sulfoxide (DMSO)-treated cells showed localization of PRRSV to the cytoplasm with a puncta shape (Fig 1D, left), whereas rottlerin-treated cells showed the absence of a signal in the cytoplasm (Fig 1D, middle), similar to EIPA treated cells (Fig 1D, right). This indicates that rottlerin treatment impaired PRRSV endocytosis. These results are consistent with our previous findings showing that rottlerin treatment suppressed PRRSV replication in cells and pigs by inhibiting PRRSV endocytosis [11].

### Rottlerin inhibited macropinocytosis of PRRSV by decreasing actin polymerization

Due to the inhibition of PRRSV replication and endocytosis by EIPA treatment (Fig 1), a selective macropinocytosis inhibitor [22], we investigated whether PRRSV entry occurs via macropinocytosis and if rottlerin inhibits PRRSV macropinocytosis. MARC-145 cells were incubated with fluorescence-labeled CF®640R dextran (10,000 MW) for an hour, which is known to enter cells via macropinocytosis [23,24] (Fig 2A). In DMSO-treated cells, dextran signals were observed in the cytoplasm, indicating the occurrence of macropinocytosis of dextran (Fig 2A, top). In contrast, rottlerin or EIPA-treated

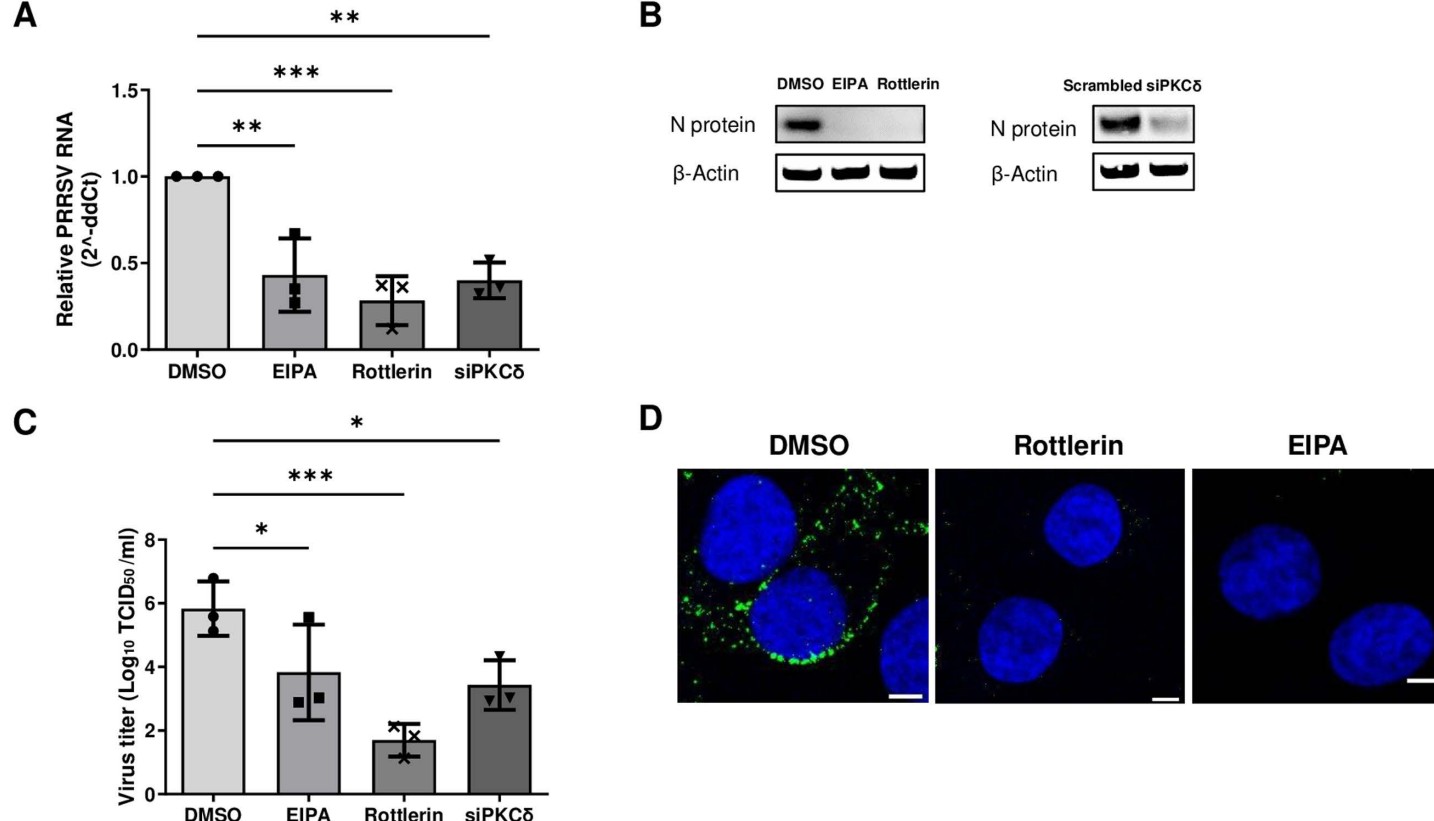

**Fig 1. Both rottlerin and PKC δ silencing inhibit PRRSV cellular entry.** (A–C) MARC-145 cells were silenced with scrambled or PKCδ-targeting siRNA, or treated with rottlerin (5 μM), or EIPA (50 μM), inoculated with PRRSV FL12 (MOI = 1), and incubated for 48 hours at 37 °C in 5% $CO_2$. (A and C) Virus titers in cell supernatants were quantified by qRT-PCR (A) or virus titration (C). (B) Cells were washed with PBS, lysed, subjected to SDS-PAGE, and transferred onto a nitrocellulose membrane. The blots were incubated with mouse anti-N monoclonal antibodies followed by goat anti-mouse IgG-HRP or mouse anti β-actin-HRP. (D) Semi-confluent MARC-145 cells were pre-treated with DMSO, rottlerin (5 μM), or EIPA (50 μM) at 37°C for 2 hours, inoculated with PRRSV FL12 (MOI = 50), and incubated for 1 hour. Cells were washed and treated with proteinase K to remove extracellular viruses for 45 minutes at 4°C. After fixation and permeabilization, the cells were stained with anti-N monoclonal antibodies, followed by Alexa Fluor 488 conjugated goat anti-mouse IgG at 37 °C. Nuclei were stained with Hoechst 33342. Statistically significant differences were measured using Kruskal–Wallis nonparametric tests and are indicated by asterisks (*, $p < 0.05$; **, $p < 0.01$). Scale bars, 5 μm. PKCδ, protein kinase C delta; PRRSV, Porcine Reproductive and Respiratory Syndrome Virus; EIPA, 5-[*N*-ethyl-*N*-isopropyl] amiloride; MOI, a multiplicity of infection; PBS, phosphate-buffered saline; SDS-PAGE, sodium dodecyl sulfate-polyacrylamide gel electrophoresis; N, nucleocapsid; HRP, horseradish peroxidase; qRT-PCR, quantitative reverse transcription–polymerase chain reaction; DMSO, dimethyl sulfoxide.

cells showed reduced dextran signals around the nucleus (Fig 2A, middle and bottom), indicating that rottlerin also inhibits macropinocytosis.

Next, PRRSV and dextran were co-inoculated into DMSO, rottlerin, or EIPA-treated cells. In DMSO-treated cells, signals for both dextran and PRRSV were observed around the nucleus with some colocalization (Fig 2B, top row). This suggests that during the endocytosis of the virus and dextran, some of them enter together through macropinocytosis. In contrast, either rottlerin or EIPA treatment significantly decreased the number of both signals in cells with significantly less colocalization between them (Fig 2C, Pearson's coefficient: DMSO, 0.43 vs. rottlerin, 0.07). These results suggest that rottlerin inhibited PRRSV macropinocytosis in MARC-145 cells.

Among the several characteristics of macropinocytosis, actin polymerization is a key factor for macropinocytosis-specific uptake [10,25]. Therefore, we tested whether rottlerin reduced actin polymerization during PRRSV

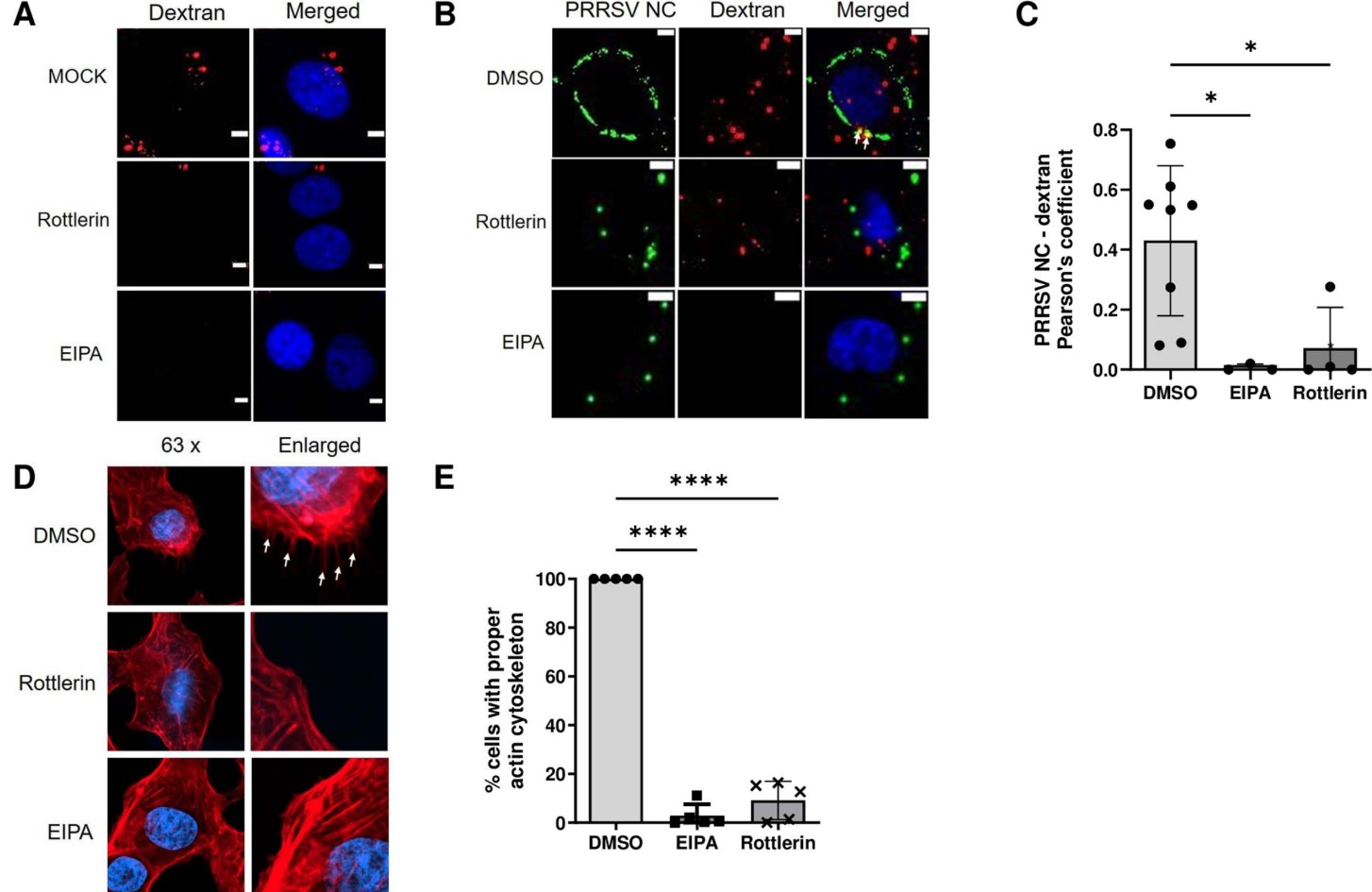

**Fig 2. PRRSV enters MARC-145 cells via macropinocytosis and rottlerin blocks the macropinocytosis by decreasing actin polymerization.**
(A-C) Semi-confluent MARC-145 cells were pre-treated with DMSO, rottlerin (5 μM), or EIPA (50 μM) for 2 hours at 37 °C. (A) The cells were washed and then inoculated with CF®640R dextran (10,000 MW) for 1 hour. Nuclei were stained with Hoechst 33342. (B-C) DMSO-, rottlerin (5 μM)-, or EIPA (50 μM)-pre-treated MARC-145 cells were washed and then inoculated with PRRSV FL12 (MOI=50) and CF®640R dextran (10,000 MW) for 1 hour. Cells were washed with sodium acetate buffer (pH 5.2) to remove extracellular signals. After fixation and permeabilization, cells were probed with anti-N monoclonal antibodies, followed by Alexa Fluor 488 conjugated goat anti-mouse IgG at 37 °C. Nuclei were stained with Hoechst 33342. Scale bars, 5 μm. (C) Co-localization of signals between dextran and PRRSV N protein in panel B was plotted by Pearson's coefficient. (D) DMSO, rottlerin (5 μM), or EIPA (50 μM) pre-treated MARC-145 cells were inoculated with PRRSV FL12 (MOI=50) and incubated with the indicated drug for 1 hour. After fixation, cells were probed with CF®640R phalloidin at 37 °C. Nuclei were stained with Hoechst 33342. Pseudopods resulting from actin rearrangement are marked as white arrows. (E) Cells with clear pseudopods were counted as positive, whereas cells without pseudopods or with small rare pseudopods were negative. Statistically significant differences are indicated by asterisks (*, *p*-value <0.05; **, *p*-value <0.01). DMSO, dimethyl sulfoxide; EIPA, 5-[*N*-ethyl-*N*-isopropyl] amiloride; PRRSV, Porcine Reproductive and Respiratory Syndrome Virus; MOI, multiplicity of infection; N, nucleocapsid.

macropinocytosis similar to EIPA. Actin dynamics were observed using fluorescently labeled phalloidin, a specific ligand of F-actin. High resolution confocal microscope revealed that PRRSV infection induced the formation of multiple pseudopods projecting from the plasma membrane into the extracellular space in DMSO-treated cells (white arrow) (Fig 2D, top row). Interestingly, rottlerin or EIPA treatment caused the disappearance of these projections (Fig 2D, middle and bottom rows). Quantitative analysis showed that cells with proper projections were significantly decreased when treated with rottlerin (*p*-value<0.05) and EIPA (*p*-value<0.01) compared with the DMSO-treated cells (Fig 2E), confirming that rottlerin inhibited PRRSV macropinocytosis by reducing actin polymerization.

## Rottlerin inhibits PRRSV endocytosis through the PKCδ-cofilin signaling pathway

PKCδ regulates actin dynamics during macropinocytosis via downstream effectors such as cofilin, slingshot phosphatase homolog 1, and LIMK1[13]. Cofilin regulates the dynamics of the actin cytoskeleton by binding to an actin filament [26,27]. LIMK1 directly phosphorylates cofilin at Ser-3, which inactivates it [13,28,29]. To evaluate the involvement of cofilin and LIMK1 in PRRSV macropinocytosis, we measured PRRSV replication in cofilin- or LIMK1-silenced MARC-145 cells. Transfection of siRNAs targeting cofilin or LIMK1 effectively knocked down the expression of each protein in the cells (Fig 3A). Interestingly silencing LIMK1 resulted in decreased expression of cofilin.

Next, we investigated the effect of cofilin or LIMK1 silencing on PRRSV replication by measuring viral titer and N protein expression in siRNA-treated cells after PRRSV infection. The N protein blot showed that silencing LIMK1 significantly inhibited N protein synthesis with the relative intensity of the band being 0.36 compared to scrambled-siRNA treated cells (Fig 3A, Relative N protein intensity:1.00). Silencing Cofilin resulted in an intensity of 0.84 for the N protein band, but this effect was not as significant as that observed with LIMK1 silencing. Consistent with the western blot results, the silencing of LIMK1 notably reduced the virus titer by 11-fold compared to scrambled-siRNA treated cells (Fig 3B). Cofilin silencing also significantly decreased the virus titer but less than that of LIMK1 silencing. These results suggest that both LIMK1 and cofilin serves as a host dependency factor essential for PRRSV infection.

To further investigate how macropinocytosis is regulated by rottlerin following PRRSV infection, we monitored the activity of cofilin or LIMK1 in PRRSV-infected cells. Since the activities of LIMK1 and cofilin are regulated by phosphorylation, we examined the levels of phosphorylated LIMK1 and cofilin (P-LIMK1 and P-cofilin) as well as total LIMK1 and cofilin using immunoblots. As shown in Fig 3C and D, the levels of P-LIMK1 after PRRSV infection with DMSO transiently increased immediately after PRRSV infection (15 mpi), slightly dephosphorylated (30 mpi), and then re-phosphorylated at a later time point (120 mpi). In contrast, rottlerin treatment rapidly attenuated the virus-induced P-LIMK1, and the P-LIMK1 levels in rottlerin-treated cells were returned similar to Mock-treated cells at 30, and 120 mpi (Fig 3C and D). Similar levels of total LIMK1 after DMSO or rottlerin treatment suggest that the changes in P- LIMK1 were not due to a change in total LIMK1. These results suggest that PRRSV infection mediates transient activation of LIMK1 at an early stage of infection through PKCδ, which was rapidly attenuated by rottlerin.

PRRSV infection gradually decreased P-cofilin levels in DMSO-treated cells, and a low level of P-cofilin was maintained up to 120 mpi (Fig 3E and F). In contrast, rottlerin treatment led to increase P-cofilin. The levels of P-cofilin in rottlerin-treated cells began to increase at 15 mpi and continued to rise until 120 mpi (Fig 3E and F). Quantitative analysis of the blot bands showed that rottlerin treatment significantly increased P-cofilin levels at 90 and 120 mpi compared with DMSO treatment (Fig 3F). These results demonstrate that PRRSV infection decreases P-cofilin levels, whereas rottlerin treatment reverses this effect. Overall, the findings suggest that rottlerin regulates LIMK1 and cofilin signaling pathways, which may contribute to the mechanism by which rottlerin inhibits PRRSV-induced macropinocytosis.

## Discussion

In this study, our microscopic and biochemical analyses demonstrated that rottlerin treatment inhibited PRRSV replication by decreasing viral endocytosis in vitro, which resulted in the suppression of viral RNA synthesis, N protein expression, and infectious virus production (Fig 1). We also showed that PRRSV entered MARC-145 cells via macropinocytosis, and rottlerin blocked it by decreasing actin polymerization (Fig 2). LIMK1 acts as a host dependency factor [30] crucial for PRRSV infection [31]. Rottlerin reversed PRRSV-induced activation of cofilin (Fig 3). These findings provide a possible explanation for one of the antiviral mechanisms of rottlerin at the early stages of virus infection, i.e., rottlerin blocks macropinocytosis of viruses by decreasing actin polymerization via the PKCδ-cofilin signaling pathway. In the case of HIV1 and Zika virus, rottlerin inhibits an early stage of their replication [32,33] and they are known to enter cells via actin-related entry pathways, indicating that rottlerin could inhibit their entry through a similar mechanism [34–36].

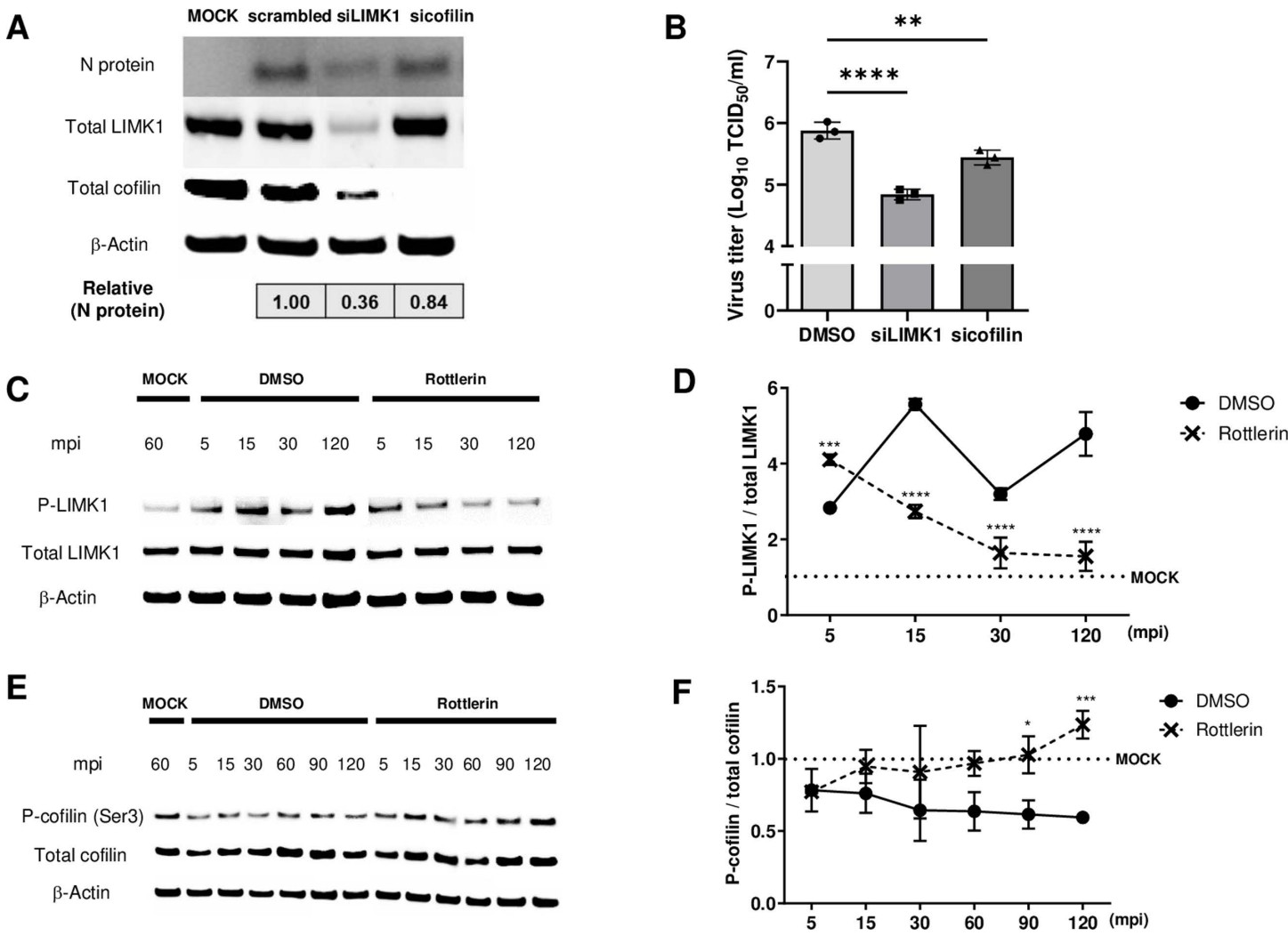

**Fig 3. Rottlerin reverses PRRSV-induced activation of cofilin and LIMK1.** (A and B) cofilin or LIMK1 was knocked down using siRNA targeting each protein. Then, cells were inoculated with PRRSV FL12 (MOI = 1) for 1 hour and vigorously washed with PBS. After 2 days of incubation, cell lysates were subjected to SDS-PAGE, transferred onto nitrocellulose membrane, and blotted for mouse anti-cofilin, rabbit anti-LIMK1, or mouse anti β-actin-HRP, followed by goat anti-rabbit IgG-HRP or goat anti-mouse IgG-HRP. (B) Viral levels in cell supernatants of the cells were quantified by virus titration. (C-F) Confluent MARC-145 cells were inoculated with PRRSV FL12 (MOI = 1) and incubated with rottlerin (5 μM) for the indicated time period at 37 °C in 5% $CO_2$. After washing with PBS, cell lysates were subjected to SDS-PAGE and transferred onto a nitrocellulose membrane. The membranes were incubated with rabbit anti-cofilin (P-cofilin, Ser3), mouse anti-cofilin (total cofilin), rabbit anti-LIMK1/2 (P-LIMK1, Thr508/505), rabbit anti-LIMK1 (total LIMK1), or mouse anti-β-actin-HRP, followed by goat anti-rabbit IgG-HRP or goat anti-mouse IgG-HRP. (D and F) Densitometry analysis plots of the kinetics of the relative levels of P-LIMK1/total LIMK1 (D) or P-cofilin/total cofilin (F) after being normalized by β-actin. PRRSV, Porcine Reproductive and Respiratory Syndrome Virus; LIMK1, LIM domain kinase 1; PBS, phosphate-buffered saline; SDS-PAGE, sodium dodecyl sulfate-polyacrylamide gel electrophoresis; HRP, horseradish peroxidase; P-cofilin, phospho-cofilin; P-LIMK1, phospho-LIMK1; siRNA, small interfering RNA; MOI, multiplicity of infection; mpi, minute post-infection.

Our results demonstrated that macropinocytosis serves as an alternative route for PRRSV to infect host cells. Our findings match with a recent study that PRRSV infects host cells through macropinocytosis, involving T-cell immunoglobulin and mucin domain proteins as well as CD163 [10]. We further hypothesize that PRRSV possibly enters cells via at least two distinct pathways: clathrin-mediated endocytosis and macropinocytosis. Few in vitro assay results support the hypothesis that a single treatment with chlorpromazine, a clathrin lattice polymerization inhibitor, or rottlerin partially

blocked PRRSV replication (Fig 1C), but the combination of both inhibitors almost completely inhibited PRRSV replication [10]. Understanding the potential PRRSV entry routes is crucial for the clinical application of antiviral drugs. To achieve an effective anti-PRRSV treatment, a combination of inhibitors targeting both entry routes, rather than a single treatment with either route inhibitor, is necessary.

Our results clarify how PRRSV induces macropinocytosis via the PKCδ-LIMK1-cofilin signaling pathway to infect host cells. Although macropinocytosis signaling mechanisms have been extensively studied over the last decade [2,13], the PRRSV-mediated signaling pathway of macropinocytosis remains unclear. Generally, to induce macropinocytosis by cofilin and LIMK1, the binding of the active form of cofilin (dephosphorylated at Ser3) depolymerizes actin by severing the actin filaments, which finally induces lamellipodium formation and macropinocytosis [37]. However, filament severing also results in actin polymerization, depending on the relative amount of actin monomers and free barb ends [38]. LIM kinases directly phosphorylate cofilin, which allows the precise regulation of actin dynamics [39]. LIMK1 is known to not only phosphorylate cofilin but also directly bind F-actin to induce actin depolymerization and membrane ruffling [40]. In the case of PRRSV, we propose that viral infection activates cofilin to increase lamellipodium formation for macropinocytosis, which increases viral uptake from the extracellular space [13]. Rottlerin treatment inhibits this activation by blocking upstream PKCδ [11]. Our microscopic data is in line with this model (Fig 2).

Our results regarding LIMK1 phosphorylation cannot be fully explained by the standard cofilin-LIMK1 pathway. In our study, PRRSV infection induced LIMK1 phosphorylation (Fig 3C and D), which stands in contrast to the standard pathway and our microscopic and cofilin phosphorylation findings, as phosphorylated LIMK1 typically phosphorylates cofilin, leading to its inactivation and the stabilization of actin filaments. Considering all our results, LIMK1 may play a noncanonical role in this virus entry process. It is possible that LIMK1 not only phosphorylates cofilin but also precisely regulates this pathway. Previous research shows LIMK1 is involved in the microtubule destabilization [41], suggesting LIMK1 may directly coordinate microtubules and actin cytoskeleton. Another research also supports this notion, indicating that pharmacological inhibition of LIMK1 does not stimulate macropinocytosis [13]. Even if LIMK1 did not follow the standard pathway, it appears that LIMK1 is a key regulator employed by PRRSV to control macropinocytosis. These results are consistent with previous studies showing that PRRSV replication requires PKCδ [42]. HIV binding to CD4＋T cells initiates transient activation of LIMK and actin polymerization [43]. Small-molecule inhibitors of LIM kinases have been reported to inhibit HIV [30]. Further research is needed to study the mechanisms underlying virus-induced macropinocytosis.

Rottlerin is a secondary metabolite isolated from *Mallotus philippinensis*, which has been traditionally used to treat tapeworm, scabies, and herpetic ringworm infections in India for several centuries [44]. Rottlerin is a selective inhibitor of PKCδ [45,46], but recent studies have reported that rottlerin inhibits multiple kinases [47–49]. In our study, PKCδ inhibition of rottlerin is highly likely to be a major blocking mechanism of PRRSV macropinocytosis, rather than other mechanisms, because silencing of PKCδ decreased PRRSV replication (Fig 1), and we have previously shown that rottlerin treatment dephosphorylates PKCδ induced by PRRSV infection [11]. Interestingly, several other studies have demonstrated the PKCδ-independent antiviral effects of rottlerin [48,49]. Ojha et al. recently showed that rottlerin inhibits La Crosse virus trafficking from the Golgi apparatus to the trans-Golgi vesicles, which may be related to decreased cellular ATP levels induced by rottlerin [48,50]. These results suggest that rottlerin can block various steps of viral replication, thus strengthening its antiviral potency. Thus, an intensive mechanistic study of the antiviral effects of rottlerin against various viruses is required.

In conclusion, the current study demonstrates that PRRSV utilizes macropinocytosis to infect host cells. Rottlerin blocks macropinocytosis of PRRSV related to PKCδ-cofilin signaling pathways by reducing actin polymerization. Our results suggest that rottlerin is a potential antiviral drug targeting macropinocytosis and its associated signaling molecules.

## Supporting information

**S1 File. Uncropped western blot images.**
(PDF)

## Acknowledgments

We are deeply grateful to the KU Research Center for Zoonosis for providing advice on the design and progress of this study.

## Author contributions

**Conceptualization:** Yeonglim Kang, Joong-Bok Lee, Seung-Yong Park, Changin Oh.

**Data curation:** Yeonglim Kang, Changin Oh.

**Formal analysis:** Yeonglim Kang.

**Funding acquisition:** Joong-Bok Lee.

**Methodology:** Yeonglim Kang.

**Resources:** Jong-Chul Choi.

**Supervision:** Changin Oh.

**Validation:** Yeonglim Kang.

**Visualization:** Yeonglim Kang.

**Writing – original draft:** Yeonglim Kang, Changin Oh.

**Writing – review & editing:** Changin Oh.

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
