## [Editor Report · Decision Letter 0]

26 Sep 2024

PONE-D-24-40479Rottlerin inhibits macropinocytosis of Porcine Reproductive and Respiratory Syndrome Virus through the PKCδ-Cofilin signaling pathwayPLOS ONE

Dear Dr. Oh,

Thank you for submitting your manuscript to PLOS ONE. After careful consideration, we feel that it has merit but does not fully meet PLOS ONE’s publication criteria as it currently stands. Therefore, we invite you to submit a revised version of the manuscript that addresses the points raised during the review process.

**ACADEMIC EDITOR: ** Dear authors, before the article can go into revision, it is necessary that you upload the full blots (including molecular weights and merges) of all the gels that make up the figures. Furthermore, excessive contrast of the bands in these figures is not recommended. Please review the manuscript and resubmit.

We look forward to receiving your revised manuscript.

Kind regards,

Gianmarco Ferrara, PhD, MVD

Academic Editor

PLOS ONE

Journal Requirements:

“This research was supported by the Korea Institute of Planning and Evaluation for Technology in Food, Agriculture, and Forestry (IPET) through the Agriculture, Food, and Rural Affairs Convergence Technologies Program for Educating the Creative Global Leader Program, funded by the Ministry of Agriculture, Food, and Rural Affairs (MAFRA) (320005-04).”

“This research was supported by the Korea Institute of Planning and Evaluation for Technology in Food, Agriculture, and Forestry (IPET) through the Agriculture, Food, and Rural Affairs Convergence Technologies Program for Educating the Creative Global Leader Program, funded by the Ministry of Agriculture, Food, and Rural Affairs (MAFRA) (320005-04). We are deeply grateful to the KU Research Center for Zoonosis for providing advice on the design and progress of this study.”

“This research was supported by the Korea Institute of Planning and Evaluation for Technology in Food, Agriculture, and Forestry (IPET) through the Agriculture, Food, and Rural Affairs Convergence Technologies Program for Educating the Creative Global Leader Program, funded by the Ministry of Agriculture, Food, and Rural Affairs (MAFRA) (320005-04).”

Additional Editor Comments (if provided):

Dear authors, before the article can go into revision, it is necessary that you upload the full blots (including molecular weights and merges) of all the gels that make up the figures. Furthermore, excessive contrast of the bands in these figures is not recommended. Please review the manuscript and resubmit.

---

## [Author Response · Author response to Decision Letter 1]

8 Nov 2024

PONE-D-24-40479

Rottlerin inhibits macropinocytosis of Porcine Reproductive and Respiratory Syndrome Virus through the PKCδ-Cofilin signaling pathway

PLOS ONE

Dear Dr. Gianmarco Ferrara,

Thank you for giving us the opportunity to submit a revised draft of the manuscript “Rottlerin inhibits macropinocytosis of Porcine Reproductive and Respiratory Syndrome Virus through the PKCδ-Cofilin signaling pathway” for publication in PLOS ONE. We appreciate the time and effort that you dedicated to providing feedback on our manuscript based on journal requirements. We have incorporated all of your suggestions. Those changes are highlighted within the manuscript. Please see below, in red, for a point-by point response to the editor’s comments.

Author response: Thank you for pointing this out. I corrected them and highlighted.

“This research was supported by the Korea Institute of Planning and Evaluation for Technology in Food, Agriculture, and Forestry (IPET) through the Agriculture, Food, and Rural Affairs Convergence Technologies Program for Educating the Creative Global Leader Program, funded by the Ministry of Agriculture, Food, and Rural Affairs (MAFRA) (320005-04).”

Author response: I revised the funding statement as follows:

“This research was supported by the Korea Institute of Planning and Evaluation for Technology in Food, Agriculture, and Forestry (IPET) through the Agriculture, Food, and Rural Affairs Convergence Technologies Program for Educating the Creative Global Leader Program, funded by the Ministry of Agriculture, Food, and Rural Affairs (MAFRA) (320005-04).” There was no additional external funding received for this study.

“This research was supported by the Korea Institute of Planning and Evaluation for Technology in Food, Agriculture, and Forestry (IPET) through the Agriculture, Food, and Rural Affairs Convergence Technologies Program for Educating the Creative Global Leader Program, funded by the Ministry of Agriculture, Food, and Rural Affairs (MAFRA) (320005-04). We are deeply grateful to the KU Research Center for Zoonosis for providing advice on the design and progress of this study.”

“This research was supported by the Korea Institute of Planning and Evaluation for Technology in Food, Agriculture, and Forestry (IPET) through the Agriculture, Food, and Rural Affairs Convergence Technologies Program for Educating the Creative Global Leader Program, funded by the Ministry of Agriculture, Food, and Rural Affairs (MAFRA) (320005-04).”

Author response: Thank you for pointing this out. I revised the Acknowledgements as follows. I revised the manuscript as well.:

We are deeply grateful to the KU Research Center for Zoonosis for providing advice on the design and progress of this study.

Author response: Thank you for pointing this out. I would like to revise my Data Availability Statement as follows:

“The minimal data set is included within the manuscript and its supporting information files.”

Author response: I have included my original uncropped and unadjusted images underlying all blot or gel results in the Supporting information file.

Additional Editor Comments (if provided):

Dear authors, before the article can go into revision, it is necessary that you upload the full blots (including molecular weights and merges) of all the gels that make up the figures. Furthermore, excessive contrast of the bands in these figures is not recommended. Please review the manuscript and resubmit.

Thank you for your consideration of our submission. We look forward to the possibility of sharing our research with your readers.

Sincerely,

Changin Oh, DVM, PhD

Associate research scientist

Department of Genetics

Yale University School of Medicine

New Haven Connecticut 06520-8005, USA

e-mail: changin.oh@yale.edu

---

## [Decision Letter · Decision Letter 1]

17 Mar 2025

PONE-D-24-40479R1Rottlerin inhibits macropinocytosis of Porcine Reproductive and Respiratory Syndrome Virus through the PKCδ-Cofilin signaling pathwayPLOS ONE

Dear Dr. Oh,

Thank you for submitting your manuscript to PLOS ONE. After careful consideration, we feel that it has merit but does not fully meet PLOS ONE’s publication criteria as it currently stands. Therefore, we invite you to submit a revised version of the manuscript that addresses the points raised during the review process.

**ACADEMIC EDITOR: **
**Based on the comment of two reviewers, your manuscript needs major revisions. **

We look forward to receiving your revised manuscript.

Kind regards,

Gianmarco Ferrara

Academic Editor

PLOS ONE

Reviewers' comments:

Reviewer's Responses to Questions

**Comments to the Author**

1. If the authors have adequately addressed your comments raised in a previous round of review and you feel that this manuscript is now acceptable for publication, you may indicate that here to bypass the “Comments to the Author” section, enter your conflict of interest statement in the “Confidential to Editor” section, and submit your "Accept" recommendation.

Reviewer #1: All comments have been addressed

Reviewer #2: All comments have been addressed

Reviewer #3: (No Response)

2. Is the manuscript technically sound, and do the data support the conclusions?

Reviewer #1: Yes

Reviewer #2: Partly

Reviewer #3: Partly

3. Has the statistical analysis been performed appropriately and rigorously? 

Reviewer #1: Yes

Reviewer #2: No

Reviewer #3: Yes

4. Have the authors made all data underlying the findings in their manuscript fully available?

Reviewer #1: Yes

Reviewer #2: Yes

Reviewer #3: Yes

5. Is the manuscript presented in an intelligible fashion and written in standard English?

Reviewer #1: Yes

Reviewer #2: Yes

Reviewer #3: Yes

6. Review Comments to the Author

Reviewer #1: Authors tried to establish the antiviral properties of Rottlerin on PRRSV and showed that the compound do have antiviral properties.

Reviewer #2: authors should deeply write the manuscript with the standard format of Plose One. There are some lacking's with introduction and materials section.

Sentence making is normal which should be more adjustable with plose one

Reviewer #3: This interesting and technically sound manuscript aims to get insight into the mechanism of cell entry of the Porcine Reproductive and Respiratory Syndrome Virus (PRRSV) to host cells and the role and mechanism of inhbition by rottlerin, a promising therapeutic agent from a natural source. However, I think that it is inconclusive and incomplete.

Major points:

1. The evidence provided for PPRV entrance by macropinocytosis (Figure 2B and C) is weak. Colocalization between dextran and PPRV is scarce (weak correlation coefficient in the control). Only one location in the image shows colocalization (two yellow dots resulting from the combination of dextran (red) and N protein (green)). The authors claim that "...virus and dextran, some of them enter together through macropinocytosis" (line 163), but it's just two dots. The experiment should be repeated or the authors should provide images of more fields. In line with this, some results are not conclusive, according to what the authors wrote: "..., suggesting macropinocytosis of dextran" (line 157).

2. Even if the authors are able to directly confirm that PPRV enters by macropinocytosis, this is just one of the possible mechanisms, because it has been clearly shown in Figure 2 that most viral particles don't. No alternative mechanism has been explored.

3. Line 429. Figure 1 legend. The title is incorrect. This experiment only tells that silencing PKC-delta, or independently adding EIPA or rottlerin inhibit uptake of the PPRV virus by MARC-145 cells, but not that rottlerin inhibit PKC-delta-mediated entry.

4. "...rottlerin inhibits multiple kinases" (line 275). Has it been demonstrated for PKC-delta? I am unable to see this in the manuscript.

Minor points:

-Please define NC in figure legends

-Please revise spacing throughout the manuscript (e.g. spaces are missing in lines 255 or 248; line 132; etc.).

-Line 220: in vitro with italics.

-Line 200: "P-cofilin" instead of "p-cofilin".

-Style: the manuscript is generally well written. However, style errors are found sometimes (e.g. lines 259 and 260, where the word cofilin is repeated unnecessarily; line 298 "...in the paper...in the paper"; lines 87 and 88).

-The definition of the SDS-PAGE abbreviation may be omitted (line 98).

-Line 144. When at for what purposes were cells treated with RIPA? Please, detail the composition of this buffer.

7. PLOS authors have the option to publish the peer review history of their article (what does this mean? ). If published, this will include your full peer review and any attached files.

**Do you want your identity to be public for this peer review?** For information about this choice, including consent withdrawal, please see our Privacy Policy .

Reviewer #1: No

Reviewer #2: No

Reviewer #3: No

---

## [Author Response · Author response to Decision Letter 2]

5 Apr 2025

PONE-D-24-40479

Rottlerin inhibits macropinocytosis of Porcine Reproductive and Respiratory Syndrome Virus through the PKCδ-Cofilin signaling pathway

PLOS ONE

Response to Reviewers

Dear Reviewers,

We would like to appreciate the reviewers for careful reading of this manuscript and for the deliberative comments and suggestions, which help to improve the quality of this manuscript. We have revised the manuscript by addressing each of the comments of the reviewers, with our responses indicated in red text. These changes are highlighted within the manuscript for clarity.

We believe that the manuscript is much improved. We hope it is now suitable for publication in PLOS ONE.

Thank you for considering this revised version.

Our response follows:

Reviewer 1

Major comments

1. The evidence provided for PPRV entrance by macropinocytosis (Figure 2B and C) is weak. Colocalization between dextran and PPRV is scarce (weak correlation coefficient in the control). Only one location in the image shows colocalization (two yellow dots resulting from the combination of dextran (red) and N protein (green)). The authors claim that "...virus and dextran, some of them enter together through macropinocytosis" (line 163), but it's just two dots. The experiment should be repeated or the authors should provide images of more fields. In line with this, some results are not conclusive, according to what the authors wrote: "..., suggesting macropinocytosis of dextran" (line 157).

Answer) We appreciate the reviewer’s perspective on our manuscript. The colocalization images presented in our study are carefully selected representative images. To ensure the robustness of our analysis, we randomly captured at least three images and quantified colocalization coefficients, as illustrated in Fig. 2C. This approach allowed us to provide comprehensive and reliable data.

Answer) We also sincerely appreciate your insightful feedback on Line 157. Dextran is a widely recognized substance that is internalized through macropinocytosis. Following a 1-hour incubation with dextran, we observed a clear dextran signal in MOCK-treated cells. In contrast, when cells were treated with EIPA, which specifically inhibits macropinocytosis, no dextran signal was detected. These results led us to conclude that dextran uptake in MOCK-treated cells occurred via macropinocytosis.

2. Even if the authors are able to directly confirm that PPRV enters by macropinocytosis, this is just one of the possible mechanisms, because it has been clearly shown in Figure 2 that most viral particles don't. No alternative mechanism has been explored.

Answer) We sincerely appreciate your insightful comments on our manuscript. As you rightly pointed out, macropinocytosis is not the only entry pathway for PRRSV; the virus also utilizes well-characterized receptors such as CD163 for endocytosis, a point we have discussed in detail in the Introduction. In this study, we clearly demonstrated that rottlerin effectively inhibits macropinocytosis, a newly recognized pathway of PRRSV entry. Further investigations are currently underway to determine whether rottlerin also affects other entry pathways of PRRSV.

3. Line 429. Figure 1 legend. The title is incorrect. This experiment only tells that silencing PKC-delta, or independently adding EIPA or rottlerin inhibit uptake of the PPRV virus by MARC-145 cells, but not that rottlerin inhibit PKC-delta-mediated entry.

Answer) We agree with the reviewer’s opinion and have revised the manuscript “Both rottlerin and PKCδ silencing inhibit PRRSV cellular entry”.

4. "...rottlerin inhibits multiple kinases" (line 275). Has it been demonstrated for PKC-delta? I am unable to see this in the manuscript.

Answer) We sincerely appreciate your valuable insight on this matter. Rottlerin is a well-established and widely utilized PKC delta-specific inhibitor (Contreras et al., 2012; Gschwendt et al., 1994; Zhao et al., 2014). Furthermore, our previous study provided clear evidence that rottlerin effectively inhibits the phosphorylation of PKC delta (doi : https://doi.org/10.1016/j.antiviral.2021.105191).

Minor comment

- Please define NC in figure legends

Answer) We fully acknowledge the reviewer’s suggestion and have accordingly changed "NC" (control siRNA) to "Scrambled" in Figure 1B. Additionally, we have changed “NC” (nucleocapsid) to “N” and provided a clear definition of it in the figure legend.

- Please revise spacing throughout the manuscript (e.g. spaces are missing in lines 255 or 248; line 132; etc.).

Answer) We fully acknowledge the reviewer’s insightful comments and have carefully revised the manuscript accordingly.

- Line 220: in vitro with italics.

Answer) We respectfully disagree with the reviewer’s suggestion for the following reasons:

The NCBI Style Guide suggests that "in vitro" and "in vivo" not be italicized (https://www.ncbi.nlm.nih.gov/books/NBK995/).

Another reference also says that common Latin terms and abbreviations such as ab initio, et al, in situ, in vitro, and in vivo should not be italicized; however, italicization should be used when referring to genus, species, subspecies, and genotypes (https://www.enago.com/academy/should-you-italicize-latin-terms-in-scientific-writing/).

- Line 200: "P-cofilin" instead of "p-cofilin".

Answer) We greatly appreciate the reviewer’s insightful feedback and have thoroughly revised the manuscript to reflect these valuable suggestions.

- Style: the manuscript is generally well written. However, style errors are found sometimes (e.g. lines 259 and 260, where the word cofilin is repeated unnecessarily; line 298 "...in the paper...in the paper"; lines 87 and 88).

Answer) We agree with the reviewer’s opinion and have revised the manuscript.

- The definition of the SDS-PAGE abbreviation may be omitted (line 98).

Answer) We appreciate the reviewer’s feedback and have made the necessary revisions to the manuscript.

- Line 144. When at for what purposes were cells treated with RIPA? Please, detail the composition of this buffer.

Answer) We sincerely appreciate your attention to detail. The correct term is EIPA, not RIPA. The manuscript has been revised accordingly.

Reviewer 2 –In the revised manuscript, there are the reviewer’s comments in “Memo” format. We have been addressed and copied each comment, followed by our corresponding responses for your convenience.

1. Line 20: “polymerization”- Please mention methodology of two lines in vitro

A. We appreciate the reviewer’s insightful comments. In response, we have provided additional explanation regarding actin polymerization and have clarified the actin staining methodology in the Methods section of revised manuscript.

2. Line 30: “(1)”- Change the line. Plagarism found

A. We greatly appreciate the reviewer’s feedback and have accordingly revised the sentence.

3. Line 47: “routes”- Write about PKCδ-Cofilin signaling pathways with references

A. We appreciate the reviewer’s comment. We have added explanation of PKCδ-Cofilin signaling pathways with a reference.

4. Line 48: “endocytosis”- Write something about rottlerin and why you chose it

A. We appreciate the reviewer’s suggestion and have included and explanation regarding rottlerin, the rationale behind selecting it for our study.

5. Line 54: “mechanism” - Introduction should make up all the aspects of the experiments along with some short description of methodology like western bloting, flurosence

A. We agree with the reviewer’s opinion. We have added a concise description of methodology used in our study.

6. Line 56: “viruses” - What do you mean by viruses, please write in methodological term

A. We agree with the reviewer’s suggestion and have replaced the term “viruses” with “Virus infection”.

7. Line 58: “MARC-145” - What it means and where you found it, write in brief

A. We appreciate the reviewer’s comment. We have added a description of MARC-145 cells with reference.

8. Line 59: “Via Fect reagent” - ???? give reference

A. We appreciated the reviewer’s feedback and have corrected the typographical error in the name of the reagent.

9. Line 60: “supernatants” - Please mention the amount in ml or liter

A. We appreciate the reviewer’s suggestion. We have included the volume of supernatants that were harvested.

10. Line 61: “Viral” - It may be virus not viral

A. We agree. We have replaced the term “viral” to “virus”.

11. Line 62: “PCR” - Write the machine name and company

A. We have added the specific name and manufacturer of the PCR machine used in our study.

12. Line 62: “titration” - How tritration is conducted, details please

A. We appreciate the reviewer’s suggestion. We have included the name of the assay along with a reference.

13. Line 68: “slide” - What type of slide it is mention here

A. We used cell culture 8-chamber slide. We have included the name and manufacturer of the slide used in our study.

14. Line 69: “preincubation” - Where the incubation was done?

A. We appreciate the reviewer’s comments. The incubation was conducted in a CO2 cell culture incubator.

15. Line 69: “rottlerin” - Do you use whole rottlerin or as a part please mention in details

A. We appreciate the reviewer’s comments and have clarified in the manuscript that full-length rottlerin was used.

16. Line 74-75: “ with or without PRRSV FL12 (MOI = 50), incubated for 1 hour, and washed with 3 M sodium acetate buffer” - You can not use with or without in methodology. Please mention the specific one

A. We agree with your suggestion. We have revised the methodology.

17. Line 80: “label” - How it labeled

A. We appreciate the reviewer’s comment. We have added detailed information about the labeling procedure to the manuscript.

18. Line 82: “or” - Mention the specific one

A. We appreciate the reviewer’s suggestion. We have specified the name of microscope used in this study.

19. Line 84: “USA” - Clear and specific writing is laking here

A. We agreed with the reviewer’s comment. We have included explanations of the microscope image processing and analysis used in this study.

20. Line 87: “LIMK1” – Elaborate

A. We agree. We have added full name of LIMK1.

21. Line 89: “Thermo Fisher Scientific - Mention the country

A. We agree. We have added the country of Thermo Fisher Scientific.

22. Line 89: “instructions” –

A. We appreciate the reviewer’s comment. We have included a brief description of the siRNA transfection method used in our study.

23. Line 94: “rottlerin (5 μM) or EIPA (50 μM)” - ????

A. We appreciate the reviewer’s comment. We have provided clarification on the treatment of each compound in the Method section.

24. Line 95: “indicated times” What do you mean by indicated time. Please specify

A. We agree with the reviewer’s suggestion. We have specified the incubation times used in the experiment.

25. Line 101: “RT” –

A. We agree. We have fixed “RT” to “room temperature”

26. Line 105: “or mouse anti-β-actin monoclonal antibody-HRP (1:50,000) (Abcam, Cambridge, UK) overnight at” - Or means specify

A. We appreciate the reviewer’s comment. We have revised the sentence to enhance clarity and readability.

27. Line 132: “ We previously reported” – What is mean??

A. We appreciate the reviewer’s feedback. The intended meaning the sentence is that while we have previously published findings on the antiviral effect of rottlerin on PRRSV, our understanding of its antiviral mechanism remains limited. We have revised the sentences to clarify this point.

28. Line 133 ~ 141: Results should be more specific. Rewrite it with specific points found in the methodology section

A. We appreciate the reviewer’s suggestion. We have revised the sentences to provide more specific details.

29. Line 151 ~166: Can not match with methodology section

A. We agree with the reviewer’s opinion regarding the unclear explanations of the dextran uptake assay. To address this, we have revised the description of the immunofluorescence assay in the Methods section to provide clearer explanations.

30. Line 219 ~224: Ref???

A. We appreciate the reviewer’s comments. We have included references to support our findings.

31. Line 242: “Our results provide insight” – Poor write up

A. We agree with the reviewer’s comment. We have revised the sentence.

32. Line 258: “which is contrary to what would be expected” – ????

A. We appreciate for the reviewer’s comment. We have revised the sentence to convey a clearer meaning.

---

## [Decision Letter · Decision Letter 2]

28 Apr 2025

Rottlerin inhibits macropinocytosis of Porcine Reproductive and Respiratory Syndrome Virus through the PKCδ-Cofilin signaling pathway

PONE-D-24-40479R2

Dear Dr. Changin Oh,

We’re pleased to inform you that your manuscript has been judged scientifically suitable for publication and will be formally accepted for publication once it meets all outstanding technical requirements.

Kind regards,

Gianmarco Ferrara

Academic Editor

PLOS ONE

Additional Editor Comments (optional):

Reviewers' comments:

Reviewer's Responses to Questions

**Comments to the Author**

1. If the authors have adequately addressed your comments raised in a previous round of review and you feel that this manuscript is now acceptable for publication, you may indicate that here to bypass the “Comments to the Author” section, enter your conflict of interest statement in the “Confidential to Editor” section, and submit your "Accept" recommendation.

Reviewer #2: All comments have been addressed

2. Is the manuscript technically sound, and do the data support the conclusions?

Reviewer #2: Yes

3. Has the statistical analysis been performed appropriately and rigorously? 

Reviewer #2: Yes

4. Have the authors made all data underlying the findings in their manuscript fully available?

Reviewer #2: Yes

5. Is the manuscript presented in an intelligible fashion and written in standard English?

Reviewer #2: Yes

6. Review Comments to the Author

Reviewer #2: It is important to follow the ethics of the publishers and also very important to show the data with more accuracy with some newly updated software

7. PLOS authors have the option to publish the peer review history of their article (what does this mean? ). If published, this will include your full peer review and any attached files.

**Do you want your identity to be public for this peer review?** For information about this choice, including consent withdrawal, please see our Privacy Policy .

Reviewer #2: **Yes: ** Sonia Akther,

---

## [Editor Report · Acceptance letter]

PONE-D-24-40479R2

PLOS ONE

Dear Dr. Oh,

I'm pleased to inform you that your manuscript has been deemed suitable for publication in PLOS ONE. Congratulations! Your manuscript is now being handed over to our production team.

Kind regards,

on behalf of

Prof. Gianmarco Ferrara

Academic Editor

PLOS ONE